# The Age of Cyclic Dinucleotide Vaccine Adjuvants

**DOI:** 10.3390/vaccines8030453

**Published:** 2020-08-13

**Authors:** Himanshu Gogoi, Samira Mansouri, Lei Jin

**Affiliations:** Division of Pulmonary, Critical Care and Sleep Medicine, Department of Medicine, University of Florida, Gainesville, FL 32610, USA; himanshu.gogoi@medicine.ufl.edu (H.G.); Samira.mansouri@medicine.ufl.edu (S.M.)

**Keywords:** cyclic dinucleotides, vaccine adjuvants, anti-tumor immunity

## Abstract

As prophylactic vaccine adjuvants for infectious diseases, cyclic dinucleotides (CDNs) induce safe, potent, long-lasting humoral and cellular memory responses in the systemic and mucosal compartments. As therapeutic cancer vaccine adjuvants, CDNs induce potent anti-tumor immunity, including cytotoxic T cells and NK cells activation that achieve durable regression in multiple mouse models of tumors. Clinical trials are ongoing to fulfill the promise of CDNs (ClinicalTrials.gov: NCT02675439, NCT03010176, NCT03172936, and NCT03937141). However, in October 2018, the first clinical data with Merck’s CDN MK-1454 showed zero activity as a monotherapy in patients with solid tumors or lymphomas (NCT03010176). Lately, the clinical trial from Aduro’s CDN ADU-S100 monotherapy was also disappointing (NCT03172936). The emerging hurdle in CDN vaccine development calls for a timely re-evaluation of our understanding on CDN vaccine adjuvants. Here, we review the status of CDN vaccine adjuvant research, including their superior adjuvant activities, in vivo mode of action, and confounding factors that affect their efficacy in humans. Lastly, we discuss the strategies to overcome the hurdle and advance promising CDN adjuvants in humans.

## 1. Introduction

Vaccines, by no doubt, are one of the most crowning achievements of medical science. The success of Edward Jenner’s smallpox vaccine and Louis Pasteur’s anthrax and rabies vaccines demonstrated the protection from the disease without transferring the disease itself. These pioneer works lay the foundation for immunology as Edward Jenner is considered the father of immunology. Vaccine development has come a long way since then and significantly reduced disease burden worldwide.

Today, a new era of recombinant and subunit vaccines with improved reactogenicity and safety profile has emerged. However, these new generations of vaccines require the addition of an adjuvant to enhance their vaccine property. Vaccine adjuvants determine the magnitude and quality of vaccine responses while maintaining a safety profile. Since its discovery by Alexander Thomas Glenny in the 1930s, alum is the choice of an adjuvant in commercially available vaccines [1]. As a vaccine adjuvant, alum enhances antibody production but is limited to the Th2 response, which is not adequate for the protection from intracellular pathogens that require Th1 and CD8^+^ cytotoxic T cell responses. Despite intensive research on new vaccine adjuvants, very few adjuvants have been licensed for commercial use today (Table 1). Furthermore, along with the new vaccine adjuvants such as MF59, AS03, and AS04, the development of new safe, potent vaccine adjuvants that generates long-lasting and balanced vaccine responses for intracellular and extracellular pathogens is a high priority for public health.

## 2. Cyclic Dinucleotides (CDNs)—An Universal Adjuvant

Most of the available vaccines confer an humoral antibody response but are unable to generate long-lived memory T cell responses, especially CD8^+^ T cell response against viral infections. For instance, the need for the cytotoxic T lymphocyte (CTL) responses in an antiviral response is undisputable. Yet, in the current COVID-19 vaccine development, most efforts aim to generate neutralizing antibodies against the SARS-CoV-2, rather than promoting the long-lived, potential cross-protective antiviral memory CD8^+^ T cells responses. The reason mostly lies in the lack of vaccine technology, e.g., adjuvants, that triggers the memory CD8^+^ T cell response during vaccination. 

Equally important, current vaccines are administered, *i.m.*, or *i.d.* or *s.c.*, and fail to generate vaccine protections on the mucosal surface. Most infectious agents enter the body on mucosal surfaces. Effective vaccines should not merely reduce the severity of the disease but prevent people from catching the infection, achieving sterile immunity. Only the induction of mucosal vaccine protection, i.e., the generation of mucosal IgA and mucosal tissue-resident memory T cells (T_RM_), can prevent the initiation, progression, and transmission of respiratory infections. For example, the oral administered Sabin poliovirus vaccine is effective because it elicits a strong mucosal IgA response and provides intestinal immunity, the site of primary poliovirus infection. In contrast, the injected Salk polio vaccine does not produce intestinal immunity and therefore is less effective at preventing the spread of poliovirus in a population.

Cyclic dinucleotide (CDNs) including cyclic di-adenosine monophosphate (cyclic di-AMP.; CDA), cyclic di-guanosine monophosphate (cyclic di-GMP; CDG), cyclic GMP-AMP (cGAMP) are a class of bacterial and mammalian second messengers with potent immunomodulatory functions [4,5,6,7,8,9,10]. CDN adjuvanted protein subunit vaccines generated mucosal immunity and protected mice from respiratory bacterial and viral infections such as influenza [8,10], Mycobacterium tuberculosis [11], anthrax [12], Klebsiella pneumoniae [13] and Streptococcus pneumoniae [4,5], Acinetobacter baumannii [14], and methicillin-resistant Staphylococcus aureus [15]. CDNs also induce long-lasting CD8^+^ T cell immunity and are therapeutic for cancer in mice [16,17]. Thus, CDNs generate long-lasting memory humoral and cellular responses in systemic and mucosal compartments [4,5,6,7,8,9,10] that can potentially fulfill all the needs for vaccines to fight intracellular, extracellular pathogens, and cancers. 

### 2.1. CDN Adjuvants Induce Potent, Safe, and Balanced Vaccine Responses

CDNs induce potent mucosal vaccine responses comparable to the cholera toxin that is the most potent experimental mucosal adjuvant so far [5]. CDNs are a more potent activator of Th1 and Th2 immune responses than lipopolysaccharide (LPS), CpG oligonucleotides (ODN), and aluminum salt-based adjuvant in mice [6]. Lastly, CDNs showed superior adjuvant properties than the poly(I:C)/CpG adjuvanted vaccination or DEC-205-mediated dendritic cell (DCs)-targeting vaccination [18]. Notably, unlike the cholera toxin, CDG does not cause acute toxicity in mice [5,19]. Recent human clinical trials on CDNs (ClinicalTrials.gov: NCT02675439, NCT03010176, NCT03172936, NCT03937141, and NCT0414414) further established an excellent safety profile in humans. 

Superior to other vaccine adjuvants that induced biased immune responses (Table 1), CDNs produce balanced memory Th1/Th2/Th17 and cytotoxic CD8^+^ T cell [6,7,16,17]. Adjuvant alum mostly induces a Th2 cell-mediated humoral response [1]. MF59^®^, the new generation of adjuvants, which is approved in Europe, does not produce a potent Th1 cellular immune response [2,3]. The mucosal adjuvant cholera toxin B drives a more restricted Th2 response [20]. On the other hand, AS04 (alum with Toll-like receptor 4 (TLR4 ligand monophosphoryl lipid A), the only TLR ligand-based mucosal adjuvant used in the US, promotes a Th1-biased response [21]. 

Ebensen et al. first reported in 2007 that *s.c.* immunization of CDG/β-galactosidase (β-Gal) not only mounted a significant humoral response as compared to only β-Gal but also a cellular response in the spleen of immunized mice [9]. The enzyme linked immunospot (ELISPOT) analysis of spleen cells showed that CDG immunization enhanced β-Gal specific memory Th1 and Th2 cells, thereby validating that CDG can modulate both humoral and a balanced Th response in a murine model [9]. In 2010 and 2011, Libanova and Ebensen et al. showed that CDG and cyclic di-inosinemonophosphate (CDI) were potent inducers of interleukin (IL)-17a (>30,000-fold as compared to antigen alone) secreting CD4^+^ Th cells along with Th1 and Th2 responses when immunized intranasally [7,19]. The ELISPOT analysis also showed that CDG/CDI were potent inducers of IFNγ producing CD8^+^ T cells [19,22]. Together, CDNs possess a broad spectrum of vaccine adjuvant activities that are suited for protection from intracellular and extracellular pathogens [7,8,11,23].

### 2.2. CDN Adjuvants Induce Anti-Tumor Responses

Cancer-related death is the second leading cause of death. The current regimen of cancer therapies are effective in extending the patients’ life but generally fail to prevent relapse and death. Immunotherapy, by activating the immune responses and generating memory cytotoxic CD8^+^ T cells, can potentially prevent tumor metastasis, regression, and cure cancer. Cancer development is accompanied by an immunosuppressive microenvironment and T cell dysfunction and exhaustion [24,25,26]. Methods to break tumor tolerance and activate anti-tumor immunity are highly desirable. 

CDNs have a potent anti-tumor activity. Demaria et al. showed that intratumoral injection of cGAMP enhanced the anti-tumor CD8^+^ T cell response inhibiting the growth of injected tumors in mice models of melanoma and colon cancer [27]. Intratumoral injection of cGAMP in B16F10 lung metastasis induced a systemic CD8^+^ T cell response to restrict the growth of distant tumors [28]. Nicolai et al. showed that the CDA activated natural killer (NK) cells-dependent and CD8^+^ T cell-independent mechanism for tumor rejection originated from different tissue types [29]. CDA induced tumor regression even in Rag2^-/-^ mice but was strongly depleted in Rag2^-/-^ Il2rg^-/-^ mice lacking NK cells, B, and T cells [29]. Moreover, suppressing the NK cell activity by NK1.1 antibody resulted in rapid tumor growth of MC-38-B2m-/- (colorectal), B16-F10-B2m-/- (melanoma), CT26-B2m-/- (colorectal), C1498-B2m-/- (leukemic), and RMA-B2m-/- (lymphoma) tumour models [29]. The study was clinically relevant as human tumors such as Hodgkin’s lymphoma often exhibit a partial loss of MHC I thereby unable to activate a CTL response. Activating NK cell-mediated anti-tumor immunity then becomes essential.

A major hurdle in cancer immunotherapy is the "cold" tumors, which lack inherent immunogenicity reflecting in the inability of checkpoint inhibitors to attenuate tumor growth. A study by Francica et al. showed that the CDN mediated tumor necrosis factor (TNF)-α production by innate immune cells are responsible for acute tumor clearance, and blocking TNF-α inhibits tumor necrosis and clearance against "cold" tumors [30].

Lastly, CDNs effectively induce DC cross-presentation to the MHC class I molecules to produce a CTL response [18]. A potent antigen-specific CD8^+^ memory T cell response by in vivo targeting of the DEC-205 receptor in DC, thereby provided an effective approach to target viral infections [18]. Combinatorial methods such as intramuscular immunization of CDA with alum have shown significant enhancement in antigen-specific CD4^+^ and CD8^+^ T cell responses while maintaining a safety profile [31].

### 2.3. CDN Adjuvants Induce Vaccine Responses on Mucosal Surface 

The mucosal surface is a natural point of entry for many respiratory pathogens such as influenza, *Streptococcus pneumoniae*, *Mycobacterium tuberculosis*, *Staphylococcus aureus*, *B. anthracis*, coronavirus, rotavirus, etc. Moreover, immunization at one mucosal surface, particularly intranasal vaccination in mice, monkeys, and humans has generated not only a local IgA response, but also enhanced the IgA response in salivary glands, upper and lower respiratory tracts, and even in distant genital tracts, and the small and large intestines [32,33,34,35,36,37]. Furthermore, rectal immunization of mice with a non-living peptide-based vaccine effectively induced a systemic CTL response [38]. Despite its role in providing a frontline of defense, immunization via the mucosal route has not been significantly explored, which is reflected by the limited number of available mucosal vaccines against influenza (Flumist), Salmonella typhi (Vivotif), Rotavirus (Rotateq, Rotatrix), Vibrio cholerae (Dukoral, Shanchol), and adeno virus [39].

A major obstacle in the development of a mucosal vaccine is the availability of a potent mucosal adjuvant, which not only generates a local mucosal IgA response but also enhances mucosal Th and CTL responses along with a systemic reaction. The most common mucosal immunostimulatory molecule used for vaccine studies are the bacterial enterotoxins, cholera toxin, and *E. coli* heat-labile toxin. Mechanistically, these toxins form a pentameric subunit, which binds to the gangliosides (preferentially GM1), thereby facilitating the uptake. Recent studies, however, raised safety concerns. Studies found that the enterotoxins are accumulated in the olfactory epithelium nerves and the olfactory bulb that induce inflammatory responses in meninges, the olfactory nerve, and glomerular layers of the olfactory bulb promoting neuronal damage [40,41,42]. 

CDNs have received enormous interest as potential mucosal adjuvants. In 2007, Ebensen et al. first demonstrated CDNs as a mucosal adjuvant [9]. They showed that CDG/β- Gal *i.n.* immunized mice not only mounted a systemic IgG response but also generated an enhanced β-Gal specific IgA response after 42 days in lung BALF and vaginal lavage [9]. Intranasal immunization of CDG/β-Gal also enhanced the serum IgG2a and IgG1 production by 640- and 320-fold as compared to only β-Gal immunized mice [9]. The systemic cellular response was also enhanced by CDG/β-Gal immunization, as was observed with the production of interferon (IFN)-γ (2000-fold), IL-5, and IL-2 in the ex vivo recall assay compared to only β-Gal immunized mice [9]. 

CDNs also induce mucosal Th17 [7,19]. The mucosal adjuvant potential of CDNs was further strengthened when Madhun et al. showed that *i.n.* administration of a plant-based H5N1 influenza antigen induced high frequencies of multifunctional Th1 cells [8]. Two doses of the CDG adjuvanted vaccine were able to mount a very high mucosal IgA response and protected against antigenically drifted H5N1 viruses [8]. In comparison, the same vaccine when it immunized *i.m.* did not generate a Th1 response, which is critical for viral infections [8]. 

The safety profile of the vaccine adjuvant is crucial while administration avoids unwanted side effects. Histopathology studies and cytokine profiling suggested that the CDG adjuvanted vaccine also induced the production of anti-inflammatory cytokine-like IL-10, thereby maintaining a balanced pro- and anti-inflammatory cytokine profile [43]. CDG also induced the production of IL-22, thereby facilitating lung epithelium repair along with effectively clearing S. pneumoniae from the organs [43]. Thus, CDNs have shown promising and safe mucosal vaccine activities in animal models.

### 2.4. CDN Adjuvants Induce Tolerogenic Responses

Distinct from the CDN-STING-type I IFNs-immunity paradigm, Andrew Mellor’s group described a CDN-STING-type I IFNs-IDO1-tolerance signaling that mediates potent T-reg responses and suppresses inflammation [44]. They first showed that DNA nanoparticles (DNP, containing PEI and plasmid DNA lacking TLR9 ligands) activate STING-dependent IDO-1 production that induces tolerance [44]. Later, they showed that the CDG treatment delayed the experimental autoimmune encephalitis (EAE) onset and reduced disease severity [45]. They identified a new CDG-STING/type I IFNs/IDO1/Tregs pathway that suppresses CNS-specific autoimmunity [45]. Recently, the same group showed that the CDN-STING-IDO-1 activity in the tumor microenvironment (TME) promoted the growth of low antigenicity Lewis lung carcinoma (LLC) [46]. Lastly, they showed that treating pre-diabetic NOD mice with cGAMP delayed the type I diabetes (T1D) onset [47] that depends on type I IFNs. Thus, CDN may be used as immune-modulatory drugs as well. 

IFNβ (Avonex^®^, Biogen, MA, USA; Rebif^®^, Merck, Darmstadt, Germany) has been used to treat Multiple Sclerosis for over 20 years. However, the in vivo mechanism of IFNβ-induced tolerance and targeted cell population remain unknown [48]. CDN-induced IFNβ may have a similar tolerance genetic effect in vivo, particularly if CDNs are targeted to the tolerogenic DCs in vivo. 

Noteworthy, Mellor’s group found that while the cGAMP treatment delayed the T1D onset in nonobese diabetic (NOD) mice, the CDG treatment did not [47]. A further analysis showed that CDG, unlike cGAMP, did not generate type I IFNs in NOD mice, though NOD mice still produced TNF in response to CDG [47]. Lastly, they found that the NOD mice contain an N210D STING polymorphism that may explain differential type I IFNs responses by cGAMP and CDG [47]. Previous studies had similar findings on the selectivity of CDNs in activating STING variants in mice. The R231A [49] or R231H [50] change in the mouse STING rendered the variants selectively to lose the response to CDG but not cGAMP. 

Notably, both human and mouse STING genes have the N210 residue while STING genes from the rat, pig, cat, dog, ferret, and hamster have the D210 residue. CDG may lose its type I IFNs-dependent anti-tumor and tolerance-inducing function in these model organisms.

## 3. Delivering CDN Vaccine Adjuvants In Vivo

CDNs carry negative charges that may prevent them from crossing the plasma membrane to activate STING in the cytosol. Indeed, in vitro cell activation by CDNs requires transfection reagents or membrane permeabilization to deliver CDNs into the cytosol. The only CDNs that directly activate cells in culture without the need for membrane permeabilization is the synthetic CDN RpRpSS-CDA [51]. In vivo, *i.n.*, *i.p.*, *s.c.* administration of CDNs can all induce vaccine responses in mice without the need for transfection reagents [8,9,15,31,52]. These CDNs are likely taken up by DCs and phagocytes directly by pinocytosis or phagocytosis [43]. 

With the advent of nanotechnology, nanoparticle, and microparticle mediated delivery vehicle such as liposomes, emulsions, virus-like particles, biodegradable polymers, immune-stimulating complexes (ISCOM) have received tremendous attention. These new delivery methods provide a sustained release of the cargo from the matrix and improve bioavailability and dosing frequency [53,54]. Thus, nanoparticle or microparticle encapsulated CDNs have been fabricated and studied as an alternative delivery approach of CDNs in vivo [10,55,56,57] (Table 2). 

### 3.1. Encapsulated CDN Adjuvants for Infectious Diseases

In 2015, Lee et al. demonstrated that the cGAMP encapsulation within polyethyleneimine/hyaluronic acid (LH) hydrogels of sized 200 nm (*i.m.*) enhanced spleen IFNβ production by 5-fold compared to the soluble cGAMP [55]. However, the antigen-specific antibody levels of cGAMP immunized vs. cGAMP/LH hydrogel immunized mice were comparable [55]. Moreover, in 2015, Hanson M.C. et al. showed that CDG encapsulated in a PEGylated liposome enhanced its retention in the lymph node by 15-fold, which otherwise disseminated into the blood following *s.c.* immunization [58]. Incorporation of HIV GP41 antigen membrane-proximal external protein (MPER) with a CDG encapsulated liposome enhanced 5.3-fold antigen-specific Tfh cell formation as compared to CDG or liposome only immunized mice [58]. There were improved germinal center B cell production and 11-fold increased antigen-specific IgG titers [58]. Lastly, the humoral responses were more durable than vaccines administered with the TLR agonist MPLA [58].

Microparticles have the advantage of biocompatibility, tunable biodegradability, ease of synthesis, and stability at elevated temperatures. Acetylated-dextran encapsulating cGAMP microparticles enhanced type I IFNs responses by nearly 1000-fold in vitro and 50-fold in vivo [69]. cGAMP microparticles (*i.m.*) increased antigen-specific antibody titers by~100-fold, enhanced Th1 responses, and expanded germinal center B cells [69]. The encapsulated cGAMP with hemagglutinin antigen achieved nearly complete protective immunity against a lethal challenge of influenza strain A/Puerto Rico/8/1934 H1N1 (PR8) seven months post-immunization [69]. The dose of CDN adjuvants (200 ng) used was 100-fold lower than previous reports [69]. Lastly, the encapsulated cGAMP elicited no observable toxicity in animals [69].

Most recently, Wang J. et al. designed a biomimetic pulmonary surfactant encapsulating the cGAMP (PS-GAMP) Perth H3N2 vaccine [10]. The PS-GAMP adjuvanted flu vaccine (*i.n.*) provided heterotypic cross-protection against the Michigan15 H1N1 strain 30 days post-immunization in ferrets [10]. These biomimetic liposomes activated STING pathways in both alveolar macrophages and alveolar epithelial cells and enhanced the recruitment of CD11b^+^ DC and the differentiation of CD8^+^ T cell and humoral response [10]. It generated heterotypic protection against H3N2, H5N1, H7N9 viruses as early as two days after a single dose of immunization and promoted the formation of a durable CD8^+^ Trm in mice [10]. 

### 3.2. Encapsulated CDN Adjuvants for Cancer Immunotherapy

Schulz et al. demonstrated that administering the cGAMP/dextran microparticle (10 μg cGAMP) via *i.t.* or *i.v.*, *i.m.*, or *s.c.* routes significantly decreased the B16F10 myeloma tumor size in mice [57]. A 100 ng of cGAMP/dextran microparticle (*i.t.*) was sufficient to reduce the tumor size as compared to the soluble cGAMP, which had to be immunized at a 10× higher concentration to obtain the same effect [57].

Liposomes composed of a phospholipid bilayer can easily penetrate the cell membrane. Miyabe et al. synthesized a pH-sensitive synthetic lipid YSK05 for cytosolic delivery of CDG as a result of the high fusogenic property of the lipid vesicle at acidic pH [61]. In a mouse model infected with EG7-OVA cells, *s.c.* immunization of liposome/CDG (300 ng) completely inhibited tumor growth while the soluble CDG could not restrict tumor growth [61]. 

A similar study using cationic PEGylated liposome and cGAMP was evaluated against metastatic lung tumors in the B16F10 melanoma mode [64]. Intravenous injection of liposome/cGAMP in tumor-bearing mice led to over 200-fold increase of lung IFNβ [64]. Liposome/cGAMP treated mice showed a significantly reduced median tumor nodule length in the lung when compared to PBS-treated controls [64]. In comparison, free cGAMP treated mice failed to reduce the tumor length [64]. Furthermore, liposome/cGAMP formulation generated tumor-specific memory and provided nearly total protection against re-challenge even after 60 days of treatment [64].

Separately, in a B16.F10 melanoma model, cGAMP encapsulating PEG-DBP pH-responsive nanoparticles enhanced STING signaling in the tumor microenvironment (TME) [63]. A single intratumoral administration of polymersome encapsulated cGAMP increased neutrophil infiltration, reduced macrophages immunosuppressive capacity, and increased DC activation in the TME [63]. In contract, mixing cGAMP with the pre-formulated vesicle did not have any immunomodulatory effect [63]. The polymersome encapsulated cGAMP resulted in an 11-fold decrease in tumor growth [63]. The intravenous administration of polymersome encapsulated cGAMP reduced the doubling time (DT) by 50% [63]. Combinatorial administration of the formulation with an immune checkpoint blockade further reduced the DT by 40%. Importantly, the mice that received combinatorial therapy exhibited no evidence of tumor burden even after 55 days of treatment [63]. Finally, intratumoral administration of the polymersome encapsulated cGAMP did not cause any organ cytotoxicity [63].

Intratumoral injection of immunostimulants is effective in the local site. However, it often fails to generate a systemic response and is ineffective on distant tumor sites. Using a phosphatidylserine coated liposome loaded with cGAMP (NP-cGAMP) in combination with radiotherapy (IR), Liu and Y. et al. detected significantly elevated lung IFNβ, and TNFα, IFNγ, IL-6, IL-12p40 in a lung metastasis mouse mode [66]. The combinatorial therapy inhibited metastases in both the IR- and non-IR-treated lungs and caused complete regression of lung metastases in some mice [66]. This combinatorial therapy also promoted an anti-tumor memory response [66].

In summary, CDNs can be administered directly via *i.n.*, *s.c.*, or encapsulated in nano/microparticles via *i.m.*, *s.c.*, *i.n.*, *i.t.*, or *i.v.* routes. Encapsulated CDNs lower the dose (>10-fold less) needed to induce effective vaccine responses in vivo. Furthermore, encapsulated CDNs offer the advantage of long term storage upon lyophilization, which is desirable in developing countries [70,71]. However, nanoparticles are often related to unwanted cytotoxicities such as cell necrosis and apoptosis [72] that may cause safety concerns in prophylactic vaccines for the general public. Nevertheless, for cancer immunotherapy, the acute toxicity may be further harnessed for tumor disintegration following local treatment [73].

## 4. Mode of Action of CDN Adjuvants

### 4.1. Molecular Mechanism of CDN Adjuvants

In 2007, Karaolis D.K. et al. first showed the immunomodulatory property of CDG and found that it is independent of TLRs or NOD receptors [13]. In 2009, McWhirter S.M. et al. showed that CDG induced robust IFNβ production in the MyD88^-/-^ TRIF^-/-^ bone marrow derived macrophage (BMDM) [74]. These results indicated that CDNs do not require the canonical TLR signaling to stimulate the type I IFNs. Notably, McWhirter S.M. et al. showed that CDG and cytosolic DNA sensing followed a common signaling pathway [74]. In 2011, Sauer J.D. et al. demonstrated that STING^-/-^ BMDM failed to induce type I IFNs in response to CDNs [75]. Moreover, in 2011, Jin L. et al. showed that expression of STING in the RAW 264.7 cell line is essential for IRF3 activation by CDNs, and STING^-/-^ BMDM failed to produce IFNβ [76]. Late in 2011, Burdette D.L. et al. found that STING is a receptor for CDG [49]. Lastly, in 2014, Blaauboer S. et al. showed that the STING^-/-^ mice do not respond to CDNs, and intranasal administration of CDNs did not induce antibody or memory Th responses in STING^-/-^ mice [52]. These results established an essential role of STING in mediating CDN-induced immune responses (Figure 1).

McWhirter S. M. et al. demonstrated that *i.p.* immunization with CDG/(human serum albumin) HSA failed to generate an immune response in *IFR3*^-/-^/*IRF7*^-/-^ mice [74]. However, *IRF3*^-/-^ mice responded to CDG immunization [74]. IRF3 mediates STING activated type I IFNs production. The results, thus, suggested that the in vivo CDN adjuvanticity may be mediated by a STING-dependent, but IRF3-type I IFNs independent pathway (Figure 1).

In 2014, Blaauboer S. et al. showed that CDG mucosal adjuvanticity depends on TNFR1, but not type I IFNs signaling [52]. In 2019, Mansour S. et al. showed that TNFR2 signaling is critical for CDG mucosal adjuvanticity in vivo [77]. TNFR2 is activated only by membrane TNF, suggesting that both secreted and membrane TNF are required for CDG adjuvant responses. Type I IFNs induction is the hallmark function of STING. Thus, the discovery that type I IFNs are dispensable for CDN-induced antibody and memory Th induction was unexpected. Nevertheless, several subsequent studies largely substantiated this conclusion [11,23,30,58].

Mechanistically, Blaauboer S. et al. showed that CDG simultaneously activates parallel IRF3-type I IFNs and RelA-TNF signaling in BMDM and bone marrow dendritic cells (BMDCs) [52]. In 2019, Mansouri S. et al. demonstrated that intranasal administration of CDG differentially activates RelA in lung conventional dendritic cell 1 (cDC1), RelB in lung conventional dendritic cell 2 (cDC2), and both RelA and RelB in lung monocyte-derived DCs (moDCs) [77]. RelB^fl/fl^CD11c^cre^ mice are deficient in CDG-induced memory Th responses but retained CDG-induced antibody production [77]. The RelA^fl/fl^CD11C^cre^ mice lost antibody and memory Th cell induction in lung mucosa but maintained antibody production and memory Th induction in the systemic compartment [78]. These results suggested that CDNs activate RelA and RelB in vivo to generate TNF, both soluble TNF and membrane TNF, that promote antibody production and memory Th cells induction in vivo (Figure 1).

Though type I IFNs are dispensable for CDN-induced antibody production, CDN-induced anti-tumor immunity was dependent on type I IFNs produced in the tumor microenvironment [27,79,80,81,82,83]. BMDCs from WT and IFNAR^-/-^ showed a decreased cross-presentation of the SIINFEKL epitope in response to CDG/ovalbumion (OVA) [83]. Intranasal CDA immunization had an impaired CD8^+^ T cell function in IFNAR^-/-^ mice [83].

### 4.2. Cellular Mechanism of CDN Vaccine Adjuvants for Infectious Diseases

DCs bridge innate and adaptive immune responses and direct vaccine responses. In 2015, Blaauboer S. et al. showed that STING^fl/fl^CD11C^cre^ mice, which lack the CDN receptor STING in CD11C^+^ cells, including DCs, are defective in CDG-induced humoral and cellular vaccine responses [43]. Thus, CDNs are recognized by STING in DCs to generate adjuvant responses.

DCs are a functionally heterogeneous population. Blaauboer S. et al. showed that intranasal administration of CDG activates cDC1, cDC2, and moDCs in the lung [43]. Interestingly, Mansouri S. et al. showed that the activation of moDCs by CDG is indirect in vivo [77]. For the reason that is still not clear, lung moDCs did not take up intranasally administered CDG [77]. Instead, moDCs are activated by TNF from CDG-stimulated cDC2 [77]. In 2019, Mansouri S. et al. demonstrated that CDG vaccine adjuvanticity is mediated by cDC2 [77]. cDC1, though, are activated by CDG in vivo, and are largely dispensable for the humoral and memory Th2/Th17 induction [77]. moDCs, on the other hand, are critical for generating lung mucosal IgA, Tfh cells, and lung memory Th cells [77,78]. In 2020, Mansouri S. et al. showed that lung moDCs differentiated into Bcl6^+^ and Bcl6^-^ mature moDCs by secreted and transmembrane TNF from cDC2, respectively in vivo [78]. The Bcl6^+^ and Bcl6^-^ mature moDCs then drive the induction of lung CD4^+^ memory T cells and lung Tfh cells, respectively, to promote lung mucosal vaccine responses [78] (Figure 2).

The in vivo cellular mechanism of CDNs may differ depending on their delivery methods and composition during immunization. However, activating DCs is likely a common mechanism for CDNs to induce vaccine responses. For example, to enhance memory Th responses, the CDN composition needs to activate cDC2. To enhance lung mucosal IgA and lung memory Th responses, moDCs activation is a must. Lastly, to generate cytotoxic CD8^+^ T cells responses, the CDN composition needs to activate cDC1 that specializes in antigen cross-presentation.

### 4.3. Cellular Mechanism of CDN Anti-Tumor Activity

The anti-tumor activity of STING depends on type I IFNs production. However, the cellular source of type I IFNs remains debatable. Tumor-infiltrating dendritic cells (DCs) were identified as the major cellular source of IFNβ at the tumor site [84]. In addition to DCs, tumor cells themselves are induced to contribute to the production of IFN-β [84]. Similarly, STING was essential for radiation-induced adaptive immune responses, which relied on type I IFN signaling on DCs [80].

Other studies identified endothelial cells as the type I IFNs producer. Upon intratumoral cGAMP injection in the mouse melanoma model, type I IFNs were produced by endothelial cells, not DCs [27]. Similarly, endothelial cells but not DCs were the principal source of spontaneously induced type I IFNs in growing tumors [27]. Similar results were obtained using an ex vivo model of cultured human melanoma explants [27], and endothelial STING expression was correlated with enhanced T-cell infiltration and prolonged survival in human colon and breast cancer [79]. It was suggested that endothelial cells initiate spontaneous and therapeutic anti-tumor immunity of CDN [27,79].

Cancer-associated fibroblasts (CAFs) and cancer infiltrating macrophages also produce type I IFNs [85]. Following transcytosis of cytoplasm (likely containing cGAMP) from cancer cells into CAFs, the STING pathway was activated, and IFNβ1 was produced in CAFs. IFNβ then drives interferon-stimulated transcriptional programs in both cancer cells and CAFs [85]. Intratumoral administration of cGAMP recruited CD45^+^ CD11b^mid^ Ly6C^+^ cell subset to mouse tumor microenvironment of 4T1 breast cancer, squamous cell carcinomas, CT26 colon cancer, or B16F10 melanoma tissue [28]. The infiltrating cells express F4/80, MHC class II, and secreted IFNβ, as well as TNFα, T cell-recruiting chemokines, CXCL10, and CXCL1 [28].

Lastly, though local activation of tumor-specific CD8^+^ effector T cells is responsible for durable anti-tumor immunity by CDNs via type I IFNs production [62], CDNs also activate NK cells to kill CD8^+^ T cell-resistant tumors [29,86]. cGAMP administration triggered STING activation and IFNβ production in myeloid cells and B cells but not NK cells [86]. Type I IFNs then activate the NK cell to kill tumor cells [29,86]. In all, STING in non-hematopoietic cells and hematopoietic cells are needed for the maximal therapeutic efficacy of CDNs [79] (Figure 3).

### 4.4. Mechanism of CDN-Induced Immune Tolerance

The tolerogenic responses induced by DNPs and CDNs depend on STING and IFN-αβ receptor genes, but not IFNγ receptor genes [45]. CD11b^+^ DCs were identified as the only cells to produce IFNβ [44]. The CD11b^+^ DCs consist of cDC2 and moDCs. It is unknown (i) which DCs subset is responsible for the CDN tolerogenic function; (ii) the unique IFNβ signaling that induces T-regs production.

## 5. Confounding Factors in CDN Adjuvanticity in Humans

Murine studies have consistently shown promising results for CDN adjuvants in infectious diseases and cancer therapeutics. However, CDN human clinical trials have failed to yield the desired effect. Merck’s synthetic CDN MK-1454 (NCT03010176) administered intratumorally failed to generate any response against metastatic solid tumors patients (head and neck squamous cell carcinoma, triple-negative breast cancer, and anaplastic thyroid carcinoma). Aduro Biotech’s synthetic CDN, ADU-S100, also had disappointing clinical responses (weak response in only 5% of the treated patients) (NCT02675439). Two confounding factors may explain the ineffectiveness of CDNs in humans.

### 5.1. The Heterogeneity of the Human STING Gene

The CDN adjuvanticity depends on STING in vivo [52]. Unlike murine *STING*, the human *STING* gene is highly heterogeneous and shows population stratification [51,87,88,89]. Human *STING* genes have five alleles that have a population frequency above 1%. They are *R232* (WT), *H232*, *HAQ* (R71H-G230A-R293Q), *AQ* (G230A-R293Q), and *Q293*. The *HAQ-STING* is extremely popular in East Asians but very rare in Africans [88]. In contrast, *AQ* and *Q293* are only found in Africans [51]. In a recent clinical trial, Sebastian M. et al. showed that *HAQ* carriers had low anti-PPS antibodies production in response to Pneumovax^®^23 immunization (ClinicalTrials.gov: NCT02471014) [90]. The result is in line with the previous data from a *HAQ* knock-in mouse mode [51]. Lastly, Kennedy R. B. et al. demonstrated that PBMC from *H232/H232* individuals had an impaired STING-mediated innate immunity (~90% decrease of IFNα induction) to poxviruses [91]. A large human population carries these *STING* alleles [51]. For example, the *HAQ/HAQ*, *HAQ/WT*, *HAQ/H232*, and *H232/H232* humans account for ~65% of East Asians and ~30% of Caucasians [51,92]. Thus, large human populations carry *STING* alleles that impact CDN vaccine responses.

### 5.2. The Impact of Age in CDN Adjuvanticity

The median age of patients in one CDN clinical trial was 61 (ClinicalTrials. gov: NCT02675439). How age affects CDN efficacy was not well-addressed. In early 2019, Wannemuehler and M et al. showed that CDN induced serum antibody production in 20-month-old female BALB/c mice [93]. They did not examine memory T cells responses or distinguish high- and low-affinity antibodies production [93]. They immunized mice (*s.c.*) with 20 µg CDN [93], a rather high dose of CDN for mice. In late December 2019, Compans and R et al. showed that cGAMP (5 µg) adjuvanted with 1 μg of hemagglutinin (HA) (*i.d.*) did not induce antibodies or protective immunity in the 19-month-old female BALB/c mice [94]. One hundred percent aged mice immunized (*i.d.*) with cGAMP/HA died subsequently from the influenza infection [94]. In comparison, 75% of aged mice immunized with Quil-A, a saponin adjuvant, with HA antigen immunization (*i.d.*) survived from the influenza infection [94]. In 2020, Gogoi H et al. reported that CDG (5 µg, *i.n.*)-induced high-affinity, durable antibodies, and Th1/Th17 responses were severely reduced in one-year-old (~equivalent age of 42.5-year-old in humans) and two-year-old (~equivalent age of 70-year-old in humans) C57BL/6J mice [95]. Aging decreases the response to vaccination [96,97,98,99]. Thus, similar to most vaccines, CDN vaccine adjuvanticity is also negatively impacted by age.

## 6. Conclusions

CDN adjuvants induce balanced, durable humoral, cellular mucosal immune responses, and potent anti-tumor immunity that is highly desirable for vaccine protection from a broad spectrum of pathogens and cancers. STING-targeting CDNs, thus, will continue to garner attention from the community. To advance CDNs as a human vaccine and cancer adjuvants, more rigorous research to understand their mode of action in vivo are needed. For example, examine the therapeutic efficacy of CDNs in vivo using mice that have the equivalent age of human cancer patients and develop CDN compositions that may enhance their adjuvant activities in aged mice.

## 7. Future of CDN Vaccine Adjuvants

Accumulating evidence in humans [90,91,100,101] and mouse models [50,51,102] indicated that the *HAQ* and *H232* alleles are impaired in the STING function. CDN-adjuvanted vaccines aim to protect the general public from the pandemic and endemic. CDN derivatives that activate HAQ and H232 STING in vivo are highly desirable. Notably, the NOD mice, which contain an N210D of STING [47], and the R231H mouse [50] react to cGAMP but not CDG for type I IFNs production. Thus, it is possible to design a CDN that activates H232 or even HAQ STING in vivo. Alternatively, He Y. et al. recently developed a cGAMP-STINGΔTM tetramer that showed promising anti-tumor activity in vivo and activates type I IFNs in the absence of STING gene [103].

Meanwhile, for CDNs that already entered clinical trials, it is worth adopting the personalized medicine concept and analyzing the data based on *STING* genotypes or designing the study based on the patients’ *STING* genotypes and age. The majority of the human population (*R232/R232*) will get the immediate benefit of the superior and unparalleled vaccine adjuvant and anti-tumor activity of CDNs.

## Figures and Tables

**Figure 1 vaccines-08-00453-f001:**
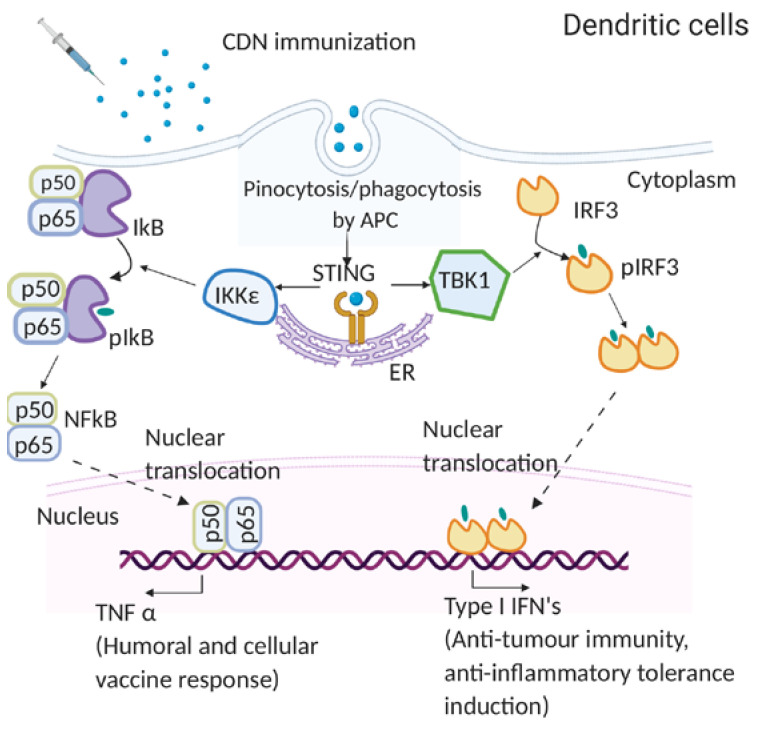
The mode of action of cyclic dinucleotides (CDNs) in dendritic cells (DCs). Immunized CDNs are taken up by pinocytosis or phagocytosis by dendritic cells in vivo. In the cytoplasm, CDNs bind to STING dimers located on the endoplasmic reticulum (ER) membrane, which undergoes conformational changes and activation. The STING (Stimulator of interferon genes) activation recruits kinases TANK binding kinase 1 (TBK 1) or IκB kinase (Iκκε). TBK 1 phosphorylates interferon regulatory factor 3 (IRF3), which dimerizes and translocates to the nucleus to activate type I IFNs. Iκκε phosphorylates nuclear factor-κB (NF-κB) inhibitor IκBα leading to dissociation of IκBα from NF-κB and translocation of the later to the nucleus to activate pro-inflammatory cytokines such as TNF-α.

**Figure 2 vaccines-08-00453-f002:**
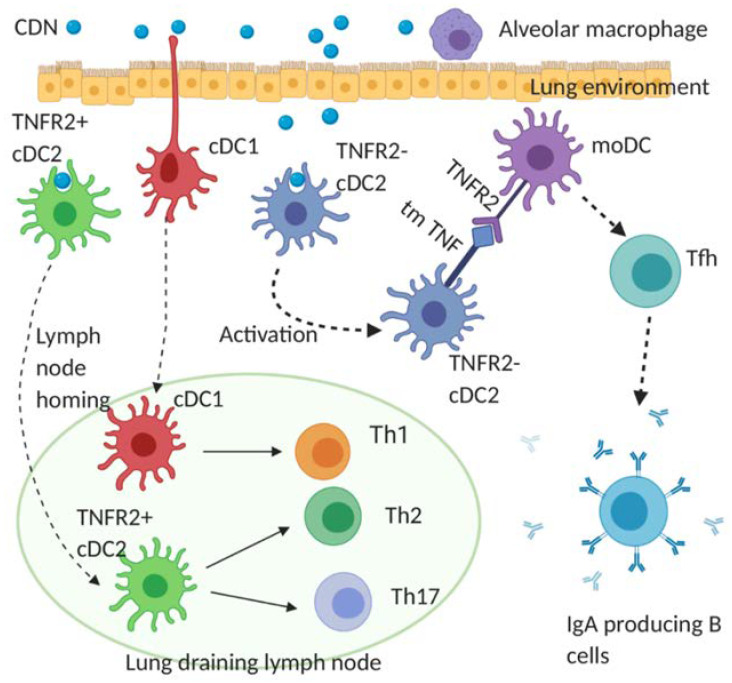
Cellular mechanism of CDN adjuvanticity by lung DCs. Intranasal immunization of CDN promotes its uptake by functionally distinct lung DC subsets: cDC1, TNFR2^+^ cDC2, and TNFR2^-^ cDC2. Upon the CDN uptake, cDC1 and TNFR2^+^ cDC2 mature and migrate towards the lung draining lymph node where they direct naïve T cells towards Th1, Th2, and Th17 effector cells. The TNFR2^-^ cDC2 population, on the CDN uptake, is activated but does not migrate. Instead, the TNFR2^-^ cDC2 produces transmembrane TNF, which engages TNFR2 on monocyte-derived DCs (moDCs) to trigger lung moDCs activation. Activated lung moDCs induce Tfh, GC formation, and IgA production in the lung.

**Figure 3 vaccines-08-00453-f003:**
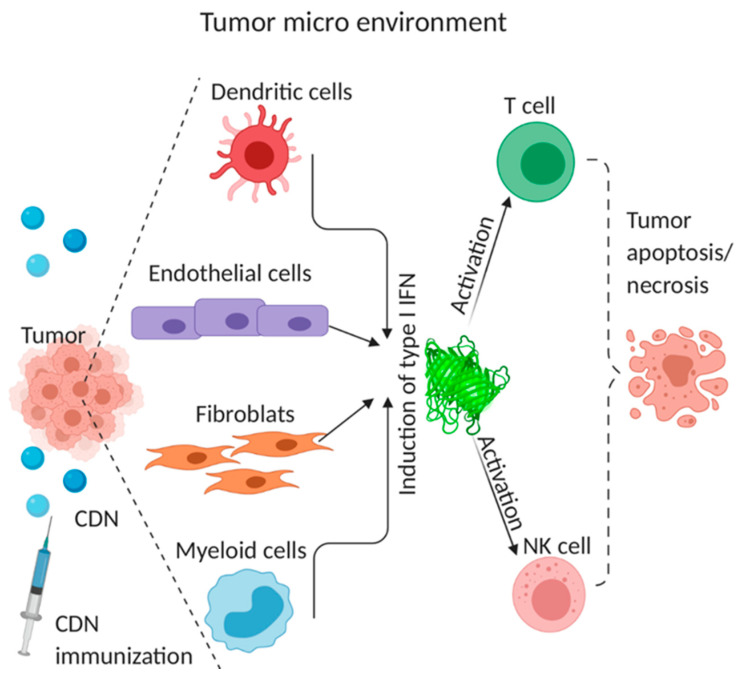
In vivo CDN-induced, type I IFNs-mediated anti-tumor immunity. The CDN immunization in the tumor microenvironment (TME) activates the STING pathway in DCs, endothelial cells, fibroblasts, and myeloid cells, leading to the production of type I IFNs. Type I IFNs act as a stimulus for DC to induce CD8^+^ T cell cross-priming. Type I IFNs also activate natural killer (NK) cells leading to NK cell-mediated cytotoxicity and tumor cell killing.

**Table 1 vaccines-08-00453-t001:** Licensed vaccine adjuvants.

Adjuvant	Composition	Immune Responses
Alum	Aluminum oxyhydroxide, Potassium aluminum sulfate [1,2,3]	Antibody, Th2 response
MF59	Squalene, polysorbate [2,3] sorbitan trioleate	Antibody, Mixed Th1/Th2 responses with Th2 biased, CD8^+^ T cell response
AS01	MPL, QS21, liposome [2,3]	Antibody, Th1 response, CD8^+^ T cell response
AS03	Squalene, polysorbate [2,3], α-tocopherol	Antibody, Th1/Th2
AS04	MPL, aluminum hydroxide [2,3]	Antibody, Th1/Th2 responses

**Table 2 vaccines-08-00453-t002:** Cyclic dinucleotide as vaccine adjuvants.

Cyclic Dinucleotide	Immune Responses
RpRp-SS-CDG/ML-RpRp-SS-cGAMP in Addvax [Squalene based oil-in-water nano emulsion] (*s.c.*) or in PBS (*i.n.*) [11]	Th1, Th17 responses; protection against *Mtb* infection
cGAMP in pulmonary surfactant mimicking liposome (*i.n.*) [10]	CD8^+^ T cell response; 100% protection against a 10LD_50_ virulent strain of CA09 H1N1
CDG in PEG lipid nanoparticles (*s.c.*) [58]	Tfh and GC B cell induction
cGAMP in ultra-pH sensitive nanoparticle ex vivo human PBMC [59]	IL-6, IL-8, G-CSF, TNF-a, type I IFNs, MIP-1α production from HIV infected PBMC, and inhibited HIV I replication
CDG in PBS (*i.n.*, *i.m. s.c.*, *i.p.*) [4,5,8,9,13,15,43]	Systemic and mucosal humoral and cellular immune responses that protect from bacterial infections
CDA with alum (*i.m.*) or in PBS (*s.c.* or *i.n.*) [7,18,31]	Balanced IgG1 and IgG2a responses; Th1/Th2/Th17 responses
(STINGel) Synthetic ML-RpRp-SS-CDA in multidomain peptide hydrogel (*s.c.*) [60]	Slow-release of CDN and highly effective in maintaining tumor clearance and rejecting tumor growth and provide durable anti-tumor immunity
CDG in YSK05 liposome (*s.c.*) [61]	Activation of antigen-specific CTL activity and restrict murine tumor growth completely
ADU-S100 (Synthetic ML-RpRp-SS-CDA) in combination with anti-PD1 or anti-CTLA4 antibody (intratumoral) [62]	Low-dosage promotes local STING activation primarily in monocytic cell lineages whereas a high-dose resulted in cellular activation at distal sites A combination of ADU-S100 and α-CPI provide durable immunity even against cold tumors
Synthetic RpRp-SS-CDG and ML-RpRp-CDA in PBAE nanoparticles and anti-PD1 antibody (intratumoral) [56]	Enhanced shelf-life stability for up to nine months. A 10-fold lower dose of PBAE/CDN nanoparticles with α-PD1 antibody resulted in significantly reduced tumor growth and reduction as compared to soluble CDN
cGAMP in acetalated dextran mps (*i.p.*, *i.m.*, *i.v.*, and intratumoral) [57]	Tumor reduction at 100-1000 fold lower dose as compared to soluble cGAMP The synergistic effect of NK cells and CD8^+^ T cells resulted in early anti-tumor efficacy in a triple-negative breast cancer model
cGAMP in pH-responsive polymersomes (intratumoral) [63]	Recruitment of active neutrophil and neutrophil chemokine *Cxcl1* in TME, the polarization of M2 macrophage towards inflammatory profile; enhanced CDN accumulation in lymph nodes and inhibited contralateral tumor growth. Increased CD8^+^ and CD4^+^ T cells with enhanced CD8^+^/CD4^+^ T cell ratio in TME
cGAMP in cationic liposomes (variable PEG levels) (*i.v.*) [64]	Durable anti-tumor immunity against a B16F10 orthotopic skin melanoma model The anti-tumor activity of cGAMP in liposome was obtained at 10-100-fold lower concentration than soluble cGAMP
CDG in cationic silica nps (intratumoral) [65]	A single intratumoral dose showed 37.5% protection and 100% protection upon tumor re-challenge Intratumoral treatment with CDG/NP in OVA expressing B16-F10 melanoma cell line induced IFNγ and TNFα producing OVA-specific CD8^+^ T cells in PBMC
ADU-S100 (Synthetic ML-RpRp-SS-CDA) in saline (intratumoral) [29,30]	NK cell-mediated and CD8^+^ T cell dispensable tumor regression and clearance in local and distal tumor sites. Increased cytokines and chemokines and innate immune cells (macrophages, neutrophils) in the TME and tumor dLNs leading to complete tumor clearance
cGAMP in phosphatidylserine liposome/radiotherapy (inhalation) [66]	Enhanced cytosolic release of cGAMP, efficient STING signaling, and type I IFN production in APCs resulting in a pro-inflammatory TME and suppressed Tregs production in TME. Liposome/cGAMP and radiation therapy-induced complete regression of lung metastasis and inhibited metastasis in both the irradiated and non-irradiated lung
STINGVAX Soluble CDA/Synthetic RR-S2-CDA/Synthetic ML-RR-S2-CDA (*s.c.*) [17]	Enhanced IFN-γ^+^ tumor-infiltrating CD8^+^T cells. STINGVAX in combination with synthetic CDA-induced type I IFN, MHC I, CD80, CD83, CD86 in monocytes and DC obtained from donor PBMCs
Synthetic DMXAA (*i.p.*, *i.v.*, intratumoral) [16,67,68]	Tumor-specific CD8^+^ T cell, and long term protection in a C1498.SIY acute leukemia model Anti-tumor immunity in distal tumor sites Macrophage repolarization in non-small cell lung cancer model

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
