# Peer review of "The Age of Cyclic Dinucleotide Vaccine Adjuvants"

_vaccines, 2020, doi:10.3390/vaccines8030453_

Round 1
Reviewer 1 Report
I recommend this manuscript for publication in Vaccines, although before proceeding I encourage authors to revise some minor issues listed below.
- Page 2, line 48: CTL should be spelled out.
- Page 2, lines 50 and 82: Please replace CD8 T cells to CD8+ T cells.
- Page 2, line 63: AMP and GMP should be spelled out.
- Page 2 lines 75, 77: LPS and DC should be spelled out.
- Throughout the manuscript, what are CDG and CDI? Please address.
- Page 3, lines 86, 88, 90, 93, 106, 109, 119: TLR, b-Gal, ELISPOT, IL, cGAMP, CDA, NK, TNF should be spelled out.
- Page 4, line 148: CDN’s is CDNs?
- Page 4, line 154: IFN should be spelled out.
- Page 5, line 184: NOD should be spelled out.
- Page 6, line 233: DC’s is DCs?
- Page 6, line 269: interleukin-6 should be deleted.
- Page 7, line 288: BMDM should be spelled out.
- Page 8, lines 309-315: The font size is different.
- Page 8, lines 317, 328, 336: BMDCs, OVA, cDC1, and cDC2 should be spelled out.
- Page 12, lines 411-423: Please check the font used.
Author Response
We appreciate the reviewer's diligence. Thank you.
Please see our point-by-point responses below.
- Page 2, line 48: CTL should be spelled out.
Response: CTL was changed to cytotoxic T lymphocyte. Please see lines 50.
- Page 2, lines 50 and 82: Please replace CD8 T cells to CD8+ T cells.
Response: Changed. Please see lines 53 and 85.
3. Page 2, line 63: AMP and GMP should be spelled out.
Response: AMP and GMP have been spelled out to adenosine mono phosphate and guanosine monophosphate. Please see line 66 and 67.
4. Page 2 lines 75, 77: LPS and DC should be spelled out.
Response: LPS and DC have been spelled to lipopolysaccharide and dendritic cell. Please see line 78 and 80.
5. Throughout the manuscript, what are CDG and CDI? Please address.
Response: CDG has been spelled out as cyclic di-GMP in page 2 line 66, CDI has been spelled out to cyclic di-inosine monophosphate in page 3 line 97.
6. Page 3, lines 86, 88, 90, 93, 106, 109, 119: TLR, b-Gal, ELISPOT, IL, cGAMP, CDA, NK, TNF should be spelled out.
Response: TLR, β-Gal, ELISPOT, IL, cGAMP, CDA, NK and TNF has been spelled out to toll like receptor (TLR, line 89), β-Gal (β-Galactoside, line 91), ELISPOT (enzyme linked immunospot, line 93), IL (interleukin, line 97), cGAMP ( cyclic GMP-AMP, line 66), CDA (cyclic di-AMP, line 63), NK (natural killer, line 113), TNF (tumor necrosis factor, line 123).
7. Page 4, line 148: CDN’s is CDNs?
Response: Changed to CDNs, line 152.
8. Page 4, line 154: IFN should be spelled out.
Response: IFN is spelled out as interferon, line 158.
9.Page 5, line 184: NOD should be spelled out.
Response: NOD is spelled out as nonobese diabetic, line 189.
10. Page 6, line 233: DC’s is DCs?
Response: Changed to DC, line 238.
11. Page 6, line 269: interleukin-6 should be deleted.
Response: Deleted, line 274.
12. Page 7, line 288: BMDM should be spelled out.
Response: BMDM was spelled out as Bone marrow derived macrophage, line 290.
13. Page 8, lines 309-315: The font size is different.
Response: Font size was corrected, line 314-320.
14. Page 8, lines 317, 328, 336: BMDCs, OVA, cDC1, and cDC2 should be spelled out.
Response: BMDCs, OVA, cDC1, and cDC2 has been spelled out to bone marrow dendritic cells (line 322), ovalbumin (line 334), conventional dendritic cell 1 (line 324), conventional dendritic cell 2 (line 324).
15. Page 12, lines 411-423: Please check the font used.
Response: Font was corrected, line 417-430.
Reviewer 2 Report
What is CD? It is not defined in the text. From the literature references I can assume it is bis-(3′,5′)-cyclic dimeric guanosine monophosphate (cdiGMP) but it is not defined in this review.
The review suggests CDNs are very good vaccine adjuvants and yet much of the review focuses on monotherapy cancer uses, perhaps a change in title is warranted and a redo of the Abstract. There is also no mention of direct comparisons of CDN to other adjuvants used commercially, and very successfully.
The abstract suggests that CDNs have superior adjuvant activities. I’m curious, what studies can be cited to support this statement?
Several other non-CDN Sting ligands are under investigation. I’m curious why these are not covered in this review?
The review stresses the great potential of CDNs but two of the most advanced compounds, MK-1454 and ADU-S100 have been very disappointing.
A table showing the structures and commercial level of CDN compounds would also provide reference for the text (similar to that done by C&EN https://cen.acs.org/articles/96/i9/STING-fever-sweeping-through-cancer.html) and some of the very pertinent references found in this review should be included here.
Line 38: AS0 formulations have demonstrated very broad-spectrum immune responses in many indications, malaria, shingles, etc. Please restate or provide references to support this statement.
There is, throughout the review a use of “s” and just general grammar mistakes that are very distracting and makes the review very difficult to read. E.g. Dinucleotides in the title would read better as Dinucleotide. many CDNs need to be CDN
Line 80: profile for file
Line 85: the adjuvant GSK has is AS04.
Section 4.1 is in italics
Author Response
We appreciate the reviewer's diligence. Thank you.
Below is our point-by-point response
What is CD? It is not defined in the text. From the literature references I can assume it is bis-(3′,5′)-cyclic dimeric guanosine monophosphate (cdiGMP) but it is not defined in this review.
Response: CDG has been spelled, line and incorporated in lines 66.
The review suggests CDNs are very good vaccine adjuvants and yet much of the review focuses on monotherapy cancer uses, perhaps a change in title is warranted and a redo of the Abstract. There is also no mention of direct comparisons of CDN to other adjuvants used commercially, and very successfully.
Response: We disagree. The review covers CDN vaccine adjuvanticity in infectious diseases, cancer, and inflammatory diseases. For example, in Section 1.2, we discussed the anti-tumor activity of CDN. In Section 1.3, we addressed CDN as vaccine adjuvants for mucosal infections. We used Section 1.4 to discuss CDN as anti-inflammatory adjuvants. In Section 2.1, we discussed CDN delivery for infectious diseases and in Section 2.2, we discussed CDN delivery for cancer immunotherapy. Similarly, we used Section 3.2 to discuss the mechanism of CDN adjuvants in infectious diseases and Section 3.3 to discuss the mechanism of CDN adjuvants in anti-cancer therapy.
As for “direct comparisons of CDN to other adjuvants used commercially” , please see table 1 and line 84-90 in Section 1.1 for the comparison of commercial adjuvants. Please see line 76-80 for the comparison of CDN adjuvanticity to alum, CpG, LPS, Poly(i:c), cholera toxin and DEC205-targeting adjuvants.
The abstract suggests that CDNs have superior adjuvant activities. I’m curious, what studies can be cited to support this statement?
Response: Please see Section 1.1, line 76-80, 84-90. References 4,5,6,13-21. Please see Section 1.2 for the discussion on the potent anti-tumor adjuvanticity of CDN. Reference 30-34. Please see Section 1.3 for CDN as an effective mucosal vaccine adjuvant. References 6,7,48.
Several other non-CDN Sting ligands are under investigation. I’m curious why these are not covered in this review?
Response: The primary focus of this article was CDN adjuvants due to which we have not discussed about non-CDN adjuvants.
The review stresses the great potential of CDNs but two of the most advanced compounds, MK-1454 and ADU-S100 have been very disappointing.
Response: In Section 4, we discussed the potential reasons leading to the failure of MK-1454 and ADU-S100, which may be attributed to STING heterogeneity in the human population, and aging.
A table showing the structures and commercial level of CDN compounds would also provide reference for the text (similar to that done by C&EN https://cen.acs.org/articles/96/i9/STING-fever-sweeping-through-cancer.html) and some of the very pertinent references found in this review should be included here.
Response: We appreciate the reviewer’s suggestion. That reference (C&EN) was published in 2018 before the disappointing trial results from MK-1454 and ADU-S100. Here, we are providing an updated, re-evaluation of CDN adjuvants citing more recent development. As for a figure of structures of commercial CDN compounds, many commercial CDN that are in clinical trials are proprietary. Their structures are not publicly available.
Line 38: AS0 formulations have demonstrated very broad-spectrum immune responses in many indications, malaria, shingles, etc. Please restate or provide references to support this statement.
Response: changed. Please see line 38-44.
There is, throughout the review a use of “s” and just general grammar mistakes that are very distracting and makes the review very difficult to read. E.g. Dinucleotides in the title would read better as Dinucleotide. many CDNs need to be CDN
Response: We have changed dinucleotides in the title to dinucleotide and CDNs to CDN in the abstract. We also change many CDNs to CDN accordingly, .
Line 80: profile for file
Response: Changed. Line 83
Line 85: the adjuvant GSK has is AS04.
Response: Changed, line 88.
Section 4.1 is in italics
Response: Change, line 419-432
Reviewer 3 Report
Stimulator of interferon genes (STING) receptor signaling pathway is to activate the induction of a cytokines and chemokines in response to environmental stimuli. Cyclic dinucleotides (CDNs) play an important role as second messengers to activate STING via NF-kB, more so in bacterial than in eukaryotes.
Various synthetic CDNs and their modified analogs have long been used as adjuvants in vaccine development and have been shown to be potent STING agonists for their use as adjuvants for vaccines. CDN's have also shown to be safe and active for mucosal vaccines in animal models.
Activation of STING pathway can trigger T cell-mediated tumor regression and help to overcome local immunosuppressive environments. They have been shown to assist eliciting both humoral and cell-mediated immune responses in animal studies using mice. Recent human studies with CDNs MK-1454 were disappointing and this review evaluates various aspects of CDN’s ability and limitation to act as vaccine adjuvant in humans.
The review attempted to discusses various ways to overcome the limitations and project the possibility of CDNs use as human vaccines in general and also for mucosal immunity relevant for COVID-19 vaccine effort.
Author Response
We appreciate the reviewer's diligence. Thank you.